# CERTIFIED ADVERSARIAL ROBUSTNESS FOR RATE ENCODED SPIKING NEURAL NETWORKS

**Bhaskar Mukhoty**[1,*], **Hilal AlQuabeh** [1,*], **Giulia De Masi**[2,3], **Huan Xiong**[1,4,†] **Bin Gu**[1,5,†]

[1] Mohamed bin Zayed University of Artificial Intelligence, UAE
[2] ARRC, Technology Innovation Institute, UAE
[3] BioRobotics Institute, Sant'Anna School of Advanced Studies, Pisa, Italy
[4] Harbin Institute of Technology, China
[5] School of Artificial Intelligence, Jilin University, China

## ABSTRACT

The spiking neural networks are inspired by the biological neurons that employ binary spikes to propagate information in the neural network. It has garnered considerable attention as the next-generation neural network, as the spiking activity simplifies the computation burden of the network to a large extent and is known for its low energy deployment enabled by specialized neuromorphic hardware. One popular technique to feed a static image to such a network is rate encoding, where each pixel is encoded into random binary spikes, following a Bernoulli distribution that uses the pixel intensity as bias. By establishing a novel connection between rate-encoding and randomized smoothing, we give the first provable robustness guarantee for spiking neural networks against adversarial perturbation of inputs bounded under $l_1$-norm. We introduce novel adversarial training algorithms for rate-encoded models that significantly improve the state-of-the-art empirical robust accuracy result. Experimental validation of the method is performed across various static image datasets, including CIFAR-10, CIFAR-100 and ImageNet-100. The code is available at https://github.com/BhaskarMukhoty/CertifiedSNN.

## 1 INTRODUCTION

Spiking neural networks (SNNs) are inspired by biological neurons that are significantly energy efficient compared to the conventional artificial neurons used in state-of-the-art deep neural networks. One of the reasons behind higher energy consumption by ordinary artificial neurons is thought to be the dense floating point dot products required between neuronal outputs (floating point) and the corresponding connection weights (floating point) to compute the input to a next-layer neuron. The artificial spiking neurons instead produce binary outputs so that the floating-point dot product can be replaced by the accumulation of edge weights that are selected by active spikes, over a fixed number of steps. Thus, the task of computing a forward-pass through a spiking neural network, when performed in specialized neuromorphic hardware (e.g., IBM TrueNorth (Merolla et al., 2014), Intel Loihi (Davies et al., 2018) etc.), requires significantly less power consumption.

Similar to the neurons in recurrent neural networks, the spiking neurons possess a temporal dimension (a.k.a. inference latency) and maintain their present state as the membrane potential, which gets updated at each time step after receiving the input in the form of weighted spikes. To process a static input, say $\mathbf{x} \in \mathbb{R}^d$, the network requires a strategy to convert it into a temporal sequence of spikes. One standard encoding technique, called constant or direct encoding, repeats the value $\mathbf{x}$, $T$ times, assuming it to be the input potential of the first-layer neurons, so that sequential spikes are generated as output of these neurons (Guo et al., 2021). Another technique called rate-encoding, inspired by

---

*First co-author: {bhaskar.mukhoty, halquabeh}@gmail.com
†Corresponding author: {huan.xiong.math, jsgubin}@gmail.com

biological evidence, intends to approximate an input $x \in [0, 1]$ with the rate of spikes (Eshraghian et al., 2021). It generates $T$ binary spikes $\{z_i\}_{i=1}^{T}$, using a Bernoulli distribution, $z_i \sim Ber(x)$, so that the average number of spikes approximates the intensity of the pixel, $\frac{1}{T} \sum_{i=1}^{T} z_i \approx x$. Yet another technique, called time-to-first-spike, generates a single spike, where the spike timing encodes the input pixel intensity (Johansson & Birznieks, 2004). It can be noted that among the encoding techniques discussed here, only rate-encoding introduces a systematic noise with the input, while the rest supply the information precisely.

As spiking neural networks aspire to replace the conventional artificial neural networks (ANN) for energy efficiency, they become exposed to intelligent attacks. Adversarial perturbation of inputs is one such attack, where an input image is altered with mild perturbation imperceptible to the human eye but can fool the classifier to make an incorrect prediction (Goodfellow et al., 2015). Defending against such attacks is imperative for many safety-critical applications. The work in (Sharmin et al., 2020) made some important experimental observations, where they found that the rate-encoded SNNs are adversarially robust compared to vanilla ANN classifiers and that the robustness of rate-encoding drops with the increase in inference latency, $T$. They further conjectured that the robustness may be due to the sparse spiking activity. However, (Kundu et al., 2021) observed that constant-encoded SNNs have lower adversarial robustness compared even to ANNs, but also have low spiking activity similar to rate-encoded SNNs. The present work is thus motivated by the need to provide a theoretical proof for the empirical observations of robustness.

We start by observing an intriguing connection between rate-encoding and the randomized smoothing framework (Lecuyer et al., 2019). Given a base classifier, the randomized smoothing technique ensures that the smooth classifier obtained as the expected classifier, under noise to the input, is adversarially robust. That is, for a given input $\mathbf{x} \in \mathbb{R}^d$, the smoothed classifier provides a radius within which any perturbation to the input will not result in a change of output. As the guarantee originates from the noise introduced to the input, it becomes a property of the smooth classifier rather than the base classifier. Nevertheless, (Salman et al., 2019) observed that it is possible to adversarially train a smooth classifier for better empirical results.

**Our Contributions:** Considering the rate-encoded network as an estimate of the smoothed classifier to Bernoulli noise enables us to establish the first certified robustness result for SNNs. The observation that rate and constant encoded classifiers correspond respectively to the smooth and base classifiers explains the superior adversarial robustness of rate encoding. The theoretical results further reveal that the certified radius suffers a drop with increasing latency, confirming the empirical observations. Table 3 reports the certified accuracies at different perturbation radii under the $l_1$-norm for various datasets on state-of-the-art architectures.

To further improve the adversarial robustness of the rate-encoded classifiers, we adopt adversarial training methods for rate-encoded classifiers. Since rate-encoding uses stochasticity to introduce Bernoulli noise, we employ the well-known Straight Through Estimator (STE) (Bengio et al., 2013; Yin et al., 2019) for back-propagation of gradients to the inputs. Experimental findings demonstrate that adversarially trained rate-encoded networks outperform their vanilla counterpart and other state-of-the-art adversarial training algorithms (Ding et al., 2022).

## 2 BACKGROUND AND RELATED WORK

**Spiking Neural Network** consists of neurons, such as Leaky Integrate and Fire (LIF) (Gerstner et al., 2014), which is governed by first-order differential equations as a continuous function of time. As the hardware required to train a network of LIF neurons operate in the discrete-time domain, the differential equations are discretized into the following recurrent equations:

$$u_i^{(l)}[t] = \beta u_i^{(l)}[t-1] + \sum_j w_{ij} x_j^{(l-1)}[t] - x_i^{(l)}[t-1] u_{th},$$

$$x_i^{(l)}[t] = h(u_i^{(l)}[t] - u_{th}) = \begin{cases} 1 & \text{if } u_i^{(l)}[t] > u_{th} \\ 0 & \text{otherwise,} \end{cases} \tag{1}$$

where, $u_i^{(l)}[t]$ denote the membrane potential of the $i$-th neuron on layer $l$ at time-step $t$, and $x_i^{(l)}[t]$ denotes the corresponding binary spike produced whenever $u_i^{(l)}[t]$ exceeds the fixed membrane

threshold $u_{th}$. The leaky constant $\beta \in (0, 1]$ determines how the membrane potential will naturally decay on each step after receiving the input spikes from previous layer neurons weighted by the connection weight $w_{ij}$.

**Adversarial Attacks:** Given a hard classifier $h : \mathbb{R}^d \to \mathcal{Y}$, where $\mathcal{Y}$ is the set of class labels, and input $\mathbf{x}$, an adversarial perturbation tries to obtain a perturbation $\boldsymbol{\delta}$, such that, $h(\mathbf{x} + \boldsymbol{\delta}) \neq h(\mathbf{x})$, under the restriction $\|\delta\|_p \leq \epsilon$. One popular way to obtain such a perturbation is to maximize the network loss over the input perturbations:

$$\max_{\|\boldsymbol{\delta}\| \leq \epsilon} \mathcal{L}(h_\theta(\mathbf{x} + \boldsymbol{\delta}), y) \tag{2}$$

The Fast Gradient Sign Method (FGSM) (Goodfellow et al., 2015) attack attempts to increase the loss using a single gradient step with a $\|\cdot\|_\infty$-constraint on $\boldsymbol{\delta}$, where the gradient step is $\boldsymbol{\delta} := \boldsymbol{\delta}_0 + \eta \nabla_\delta \mathcal{L}$, with $\boldsymbol{\delta}_0 = 0$. Applying the norm constraint on $\boldsymbol{\delta}$ and maximizing over $\eta$ we obtain,

$$\boldsymbol{\delta} := \max_\eta clip(\eta \nabla_\delta \mathcal{L}, [-\epsilon, \epsilon]) = \epsilon \cdot sign(\nabla_\delta \mathcal{L}) \tag{3}$$

Often, the optimization is performed over several gradient ascent steps with $\Pi$ as the projection operation performed on the $l_p$-norm ball of radius $\epsilon$:

$$\delta_{t+1} = \Pi_\epsilon(\boldsymbol{\delta}_t + \eta \nabla_\delta \mathcal{L}(h_\theta(\mathbf{x} + \boldsymbol{\delta}_t), y) \tag{4}$$

when the projection step is performed on a $l_\infty$-ball, the attack is popularly known as projected gradient descent (PGD) attack (Madry et al., 2018).

**Empirical Defense:** To defend a classifier against adversarial attacks, several strategies are proposed that provide empirical robustness to ANNs (Papernot et al., 2016; Guo et al., 2018). One popular defense strategy is adversarial training (Madry et al., 2018), where a classifier is trained with respect to adversarial examples instead of clean ones. As an optimization to find the worst adversarial perturbation is NP-hard, such a strategy, though empirically adequate, does not guarantee that no adversarial perturbation can exist for a given input. In the context of SNNs, to improve the empirical robustness of constant-encoded networks, (Kundu et al., 2021) proposed to perturb the input image separately at different time steps instead of supplying the same image repeatedly, thus avoiding any computational burden due to adversarial training. More recently, (Ding et al., 2022) proposed the adversarial training of constant encoded SNNs with additional regularization, called regularized adversarial training (RAT), which has been shown to provide higher robustness against adversarial attacks.

**Certifiable Defense:** Though empirical defenses are practical, they do not ensure the absence of an adversarial perturbation. In practice, the empirical defense strategies are often broken with more potent attacks. Given an input $\mathbf{x}$, a classifier $h$ is said to be provably robust under norm $l_p$ attacks if there exists a radius $r$ within which any perturbation to $\mathbf{x}$ does not change the output of the classifier, i.e.,

$$\forall \mathbf{x}' : \|\mathbf{x} - \mathbf{x}'\|_p \leq r \text{ we have, } h(\mathbf{x}') = h(\mathbf{x}) \tag{5}$$

Through bounding input/output of the activation and bound propagation, certified training strategies prove whether adversarial perturbation can exist within a particular radius of any input. A recent method that adopts such existing literature (Xu et al., 2020; Wang et al., 2021) to SNNs is known as (S-IBP, S-CROWN) (Liang et al., 2022).

**Randomized Smoothing:** One of the effective strategies to provide a probabilistic certificate of robustness in a classifier agnostic way is through randomized smoothing (Lecuyer et al., 2019), which gives provable robustness against adversarial attacks restricted under $l_1$ or $l_2$ norm. Randomized smoothing analyses the effect of adding input noise, such as Gaussian or Laplacian, to a base classifier. A classifier that gives class probabilities as output is known as a soft classifier, in contrast to a hard classifier that returns the predicted class label. Given a (soft) base classifier $f : \mathbb{R}^d \to \mathbb{P}(\mathcal{Y})$, the Gaussian smoothing constructs a smooth classifier $g : \mathbb{R}^d \to \mathbb{P}(\mathcal{Y})$:

$$g(\mathbf{x}) = \mathbb{E}_{\boldsymbol{\epsilon} \sim N(0, \sigma^2 \cdot I)}[f(\mathbf{x} + \boldsymbol{\epsilon})] = \mathbb{E}_{\mathbf{z} \sim N(\mathbf{x}, \sigma^2 \cdot I)}[f(\mathbf{z})] \tag{6}$$

Given, $a = \arg\max_{y \in \mathcal{Y}} g(\mathbf{x})_y$, $b = \arg\max_{y \neq a} g(\mathbf{x})_y$ and let $p_a, p_b$ be the corresponding probabilities with their statistical estimates, $\underline{p_a} \leq p_a$ and $p_b \leq \overline{p_b}$, we have from (Cohen et al., 2019):

$$\arg\max_{y \in \mathcal{Y}} g(\mathbf{x} + \boldsymbol{\delta})_y = a \tag{7}$$

$$\forall \boldsymbol{\delta} : \quad \|\boldsymbol{\delta}\|_2 \leq \frac{\sigma}{2}(\Phi^{-1}(\underline{p_a}) - \Phi^{-1}(\overline{p_b})) \tag{8}$$

where, $\Phi^{-1}$ is the inverse Gaussian CDF. This implies that the smooth classifier, $g$, when deployed in the test time, can tolerate any adversarial attack with a bounded norm of radius $R = \frac{\sigma}{2}(\Phi^{-1}(\underline{p_a}) - \Phi^{-1}(\overline{p_b}))$. As probabilities $p_a$ and $p_b$ are difficult to compute analytically, they are estimated using Monte-Carlo (MC) simulations (Cohen et al., 2019; Salman et al., 2019), by evaluating the base classifier on multiple noisy input, i.e.,

$$\hat{g}(\mathbf{x}) \approx \frac{1}{m} \sum_{i=1}^{m} f(\mathbf{z}_i) \quad \text{where, } \mathbf{z}_i \sim Noise(\mathbf{x}) \tag{9}$$

so that the bounds $\underline{p_a} \leq p_a$ and $p_b \leq \overline{p_b}$ hold with high probability.

# 3 ADVERSARIAL ROBUSTNESS USING BERNOULLI SMOOTHING

## 3.1 RANDOMIZED SMOOTHING VIA BERNOULLI NOISE

We propose to analyze the robustness properties of a classifier that uses Bernoulli noise as the source of randomness. The input to the smoothed classifier $g$ can be an image $\mathbf{x} \in [0,1]^d$, where individual pixel values $x_i \in [0,1]$ are bounded. For Bernoulli smoothing, the individual pixel intensity $x_i$ can be treated as the bias of a Bernoulli random variable, $z_i$,

$$z_i \in \{0,1\}: \quad z_i \sim Ber(x_i) \tag{10}$$

which in vector notation we write as, $\mathbf{z} \sim Ber(\mathbf{x})$. The theorem below proves that we can construct a provably robust classifier $g$ using Bernoulli noise, given a base classifier $f$.

**Theorem 1.** *Given a base classifier $f : [0,1]^d \to \mathbb{P}(\mathcal{Y})$, we construct a smooth classifier $g : [0,1]^d \to \mathbb{P}(\mathcal{Y})$, such that,*

$$g(\mathbf{x}) = \mathbb{E}_{\mathbf{z} \sim Ber(\mathbf{x})}[f(\mathbf{z})] \tag{11}$$

*Let $p_a = \max_{y \in \mathcal{Y}} g(\mathbf{x})_y$, $p_b = \max_{y \neq a} g(\mathbf{x})_y$, then,*

$$\forall \boldsymbol{\delta} : \|\boldsymbol{\delta}\|_1 < \frac{p_a - p_b}{2} \quad \arg\max_{y \in \mathcal{Y}} g(\mathbf{x} + \boldsymbol{\delta}) = a \tag{12}$$

As Monte-Carlo simulations estimate the probabilities $p_a$ and $p_b$, we expect to obtain bounds $p_a \geq \underline{p_a}$ and $p_b \leq \overline{p_b}$ with high probability. Since, $\underline{p_a} - \overline{p_b} \leq p_a - p_b$,

**Corollary 1.** *We have,*

$$\forall \boldsymbol{\delta} : \|\boldsymbol{\delta}\|_1 < \frac{\underline{p_a} - \overline{p_b}}{2} = \underline{p_a} - 0.5 \quad \arg\max_{y \in \mathcal{Y}} g(\mathbf{x} + \boldsymbol{\delta}) = a$$

*where in the last equality we used the upper bound, $\overline{p_b} := 1 - \underline{p_a}$.*

To prove the Theorem1, we would require to use Definition1 along with lemma 2 and 3.

**Definition 1.** *A function $h : \mathbb{R}^d \to \mathbb{R}$ is said to be L-Lipschitz w.r.t. to the norm $\|\cdot\|$, if,*

$$\forall \mathbf{x}, \mathbf{y} \in \mathbb{R}^d : |h(\mathbf{x}) - h(\mathbf{y})| \leq L\|\mathbf{x} - \mathbf{y}\|$$

*Equivalently, if $h$ is differentiable, it is L-Lipschitz if and only if,*

$$\forall \mathbf{x}, \quad \|\nabla h(\mathbf{x})\|_* \leq L$$

*where $\|\cdot\|_* : \mathbb{R}^d \to \mathbb{R}$ denotes the dual norm of $\|\cdot\| : \mathbb{R}^d \to \mathbb{R}$, and it is defined as: $\|\mathbf{y}\|_* = \sup_{\|\mathbf{x}\|=1} \mathbf{x}^T \mathbf{y}$*

**Lemma 2.** *Let $h : \mathbb{R}^d \to \mathbb{P}(\mathcal{Y})$ be soft classifier and suppose it is L-Lipschitz with respect to each of the classes $\mathbf{x} \to h(\mathbf{x})_y$ and the norm $\|\cdot\|$. For a given input $\mathbf{x}$, if we have $p_a = \max_{y \in \mathcal{Y}} h(\mathbf{x})_y$, $p_b = \max_{y \neq a} h(\mathbf{x})_y$, then,*

$$\forall \boldsymbol{\delta} : \|\boldsymbol{\delta}\| < \frac{p_a - p_b}{2L} \quad \arg\max_{y \in \mathcal{Y}} h(\mathbf{x} + \boldsymbol{\delta}) = a \tag{13}$$

A proof for the above can be found in Lemma 2.1 (Li, 2019).

**Lemma 3.** *For all $y$, $g(\mathbf{x})_y$ is 1-Lipschitz under $\|\cdot\|_1$-norm.*

The proof of this lemma is given in the appendix.

## 3.2 MULTI-BERNOULLI SMOOTHING AND ITS APPLICATION TO SNN

Armed with the proof that the smooth classifier is robust when a single Bernoulli variable is used to encode a pixel $x_i$, let us now move to the more practical situation of rate-encoding, where T-independent Bernoulli variables encode the input pixel. To fix the notation, let us consider constant encoding replicating the original input $\mathbf{x} \in \mathbb{R}^d$, $T$ times, creating temporal encoding that we denote using the notation $[\mathbf{x}]^T$. To be consistent, let us denote a base classifier $f_T : [0, 1]^d \to \mathbb{P}(\mathcal{Y})$ which accepts the input without the encoding, and further let us define, $f_T(\mathbf{x}) = \tilde{f}_T([\mathbf{x}]^T)$, where the classifier $\tilde{f}_T : [0, 1]^{T \times d} \to \mathbb{P}(\mathcal{Y})$ receives the replicated input. The notation of $\tilde{f}_T$ helps us demonstrate the definition of the corresponding smooth classifier, which receives replicated input $[\mathbf{x}]^T$ encoded with independent Bernoulli noise. The smooth classifier $g_T : [0, 1]^d \to \mathbb{P}(\mathcal{Y})$ can now be defined as:

$$g_T(\mathbf{x}) = \mathbb{E}_{\mathbf{z}_i \sim Ber^T(x_i)}[\tilde{f}_T(\langle \mathbf{z}_1, \mathbf{z}_2, \cdots, \mathbf{z}_d \rangle)] \tag{14}$$

where, the smooth classifier $g_T$ uses $T$ independent Bernoulli variables $\mathbf{z}_i = \langle z_{i,1}, z_{i,2}, \cdots, z_{i,T} \rangle \in \{0, 1\}^T$ to encode each input pixel $x_i \in [0, 1]$. The notation $Ber^T(x_i)$ describes the joint Bernoulli distribution of $T$ variables, each having the same bias $x_i$. We can now show,

**Lemma 4.** *For all $y$, $g_T(\mathbf{x})_y$ is $T3^{T-1}$-Lipschitz under $\|\cdot\|_1$-norm.*

We differ the proof to the appendix.

**Theorem 5.** *For the smooth classifier $g_T$ defined as above, and $p_a = \max_{y \in \mathcal{Y}} g_T(\mathbf{x})_y$, $p_b = \max_{y \neq a} g_T(\mathbf{x})_y$, we have:*

$$\forall \boldsymbol{\delta} : \|\boldsymbol{\delta}\|_1 < \frac{p_a - p_b}{2T \, 3^{T-1}} \quad \arg\max_{y \in \mathcal{Y}} g_T(\mathbf{x} + \boldsymbol{\delta}) = a \tag{15}$$

The proof of the theorem follows from lemma2 and 4. It is interesting to note that Theorem 1 is a special case of Theorem5 when $T = 1$.

**Corollary 2.** *Similar to corollary 1, if we estimate $p_a \geq \underline{p_a}$ with high probability, then robustness guarantee on the perturbation $\boldsymbol{\delta} : \|\boldsymbol{\delta}\|_1 < \frac{\underline{p_a} - 0.5}{T 3^{T-1}}$, holds with the same probability.*

One may highlight that the certified radius drops quickly with $T$, which, we agree, can perhaps be improved with a tighter bound on the Lipschitzness. Also, the largest $l_1$-radius on which an input can obtain a robustness guarantee is upper bounded by 0.5, with $T = 1$. To put it in perspective, it allows a change of $\frac{127}{255}$, which, assuming 8-bit encoding of pixels, allows a single pixel to be changed by 127 or 127 pixels to be changed by 1. In comparing the results with Gaussian smoothing, one may find that their guarantee can hold for arbitrary radius. However, certificates for larger radii require a significantly large number of MC simulations. Moreover, such results are impossible for discrete random variables such as Bernoulli.

## 3.3 PREDICTION AND CERTIFIED ROBUSTNESS

Estimation of the smooth classifier $g_T(\mathbf{x})$ requires MC simulations, which corresponds to the evaluation of the base classifier $f_T$ on rate-encoded input. To evaluate $g_T$, we use the predict and certify functions described in Algorithm 1 and 2, which are similar to that of (Cohen et al., 2019), with the exception that the Gaussian noise is replaced by multi-Bernoulli noise. The sub-routine RateEncode($f_T, \mathbf{x}, m$) evaluates the classifier $f_T$ after rate-encoding the input $\mathbf{x}$. The process is repeated $m$ times, and the vector $counts$ records the number of times a particular class is predicted. For proof that Algorithm 1 makes an incorrect prediction $\hat{c}_a \neq g_T(\mathbf{x})$ with a probability upper bounded $\alpha$, please refer to proposition 1 (Cohen et al., 2019) that in turn uses results from (Hung & Fithian, 2019). Algorithm 2 describes the certify function, which works in two stages. First, using $m_0$ MC simulations, it identifies the predicted class $\hat{c}_a$, then to find the lower bound $\underline{p_a}$, it uses another $m$ simulations. Proof that Algorithm 2 makes an incorrect prediction with probability upper bounded by $\alpha$ can also be found in proposition 2 (Cohen et al., 2019).

**Certified Test Accuracy:** For each input $\mathbf{x}$, the certify function either abstains or returns the predicted class with a $l_1$ radius, within which no adversarial example exists. Two conditions are verified

on each input data point to obtain the certified test accuracy of a model at a given radius $r$. First, the predicted class should match the target label $y$; second, the radius returned by the certify function should be larger or equal to $r$. The fraction of test data that satisfies both conditions without abstaining gives the certified test accuracy of a model. Observe that this is an estimate of the true certified test accuracy, as we only have an estimate of the smooth classifier $g_T$. However, the estimate can be improved by reducing the error rate $\alpha$ given Algorithms 1 and 2.

---

**Algorithm 1** predict $g_T(\mathbf{x})$

---

**Require:** $\tilde{f}_T, \mathbf{x}, m, \alpha$
$\quad counts \leftarrow \text{RateEncode}(\tilde{f}_T, \mathbf{x}, m)$
$\quad \hat{c}_a, \hat{c}_b \leftarrow \text{top two indicies in } counts$
$\quad n_a, n_b \leftarrow counts[\hat{c}_a], counts[\hat{c}_b]$
$\quad \textbf{if } \text{BinomPValue}(n_a, n_a + n_b, 0.5) \leq \alpha \textbf{ then}$
$\quad\quad \textbf{return } \hat{c}_a$
$\quad \textbf{else}$
$\quad\quad \textbf{return } \text{abstain}$
$\quad \textbf{end if}$

---

**Algorithm 2** certify $g_T$ around $\mathbf{x}$

---

**Require:** $\tilde{f}_T, \mathbf{x}, m_0, m, \alpha$
$\quad counts0 \leftarrow \text{RateEncode}(\tilde{f}_T, \mathbf{x}, m_0)$
$\quad \hat{c}_a \leftarrow \text{top index in } counts0$
$\quad counts \leftarrow \text{RateEncode}(\tilde{f}_T, \mathbf{x}, m)$
$\quad \underline{p_a} \leftarrow \text{LowerConfBound}(counts[\hat{c}_a], m, 1 - \alpha)$
$\quad \textbf{if } \underline{p_a} \geq 0.5 \textbf{ then}$
$\quad\quad \textbf{return } \text{prediction } \hat{c}_a, \text{ radius } \frac{p_a - 0.5}{T3^{T-1}}$
$\quad \textbf{else}$
$\quad\quad \textbf{return } \text{abstain}$
$\quad \textbf{end if}$

---

## 4 ADVERSARIAL TRAINING OF SNN CLASSIFIER

The notion of adversarial training can be seen as a min-max problem, where the objective is first maximized with respect to bounded input perturbations and then minimized over the model parameters.

$$\min_\theta \frac{1}{|S|} \sum_{\mathbf{x},y \in S} \max_{\|\delta\| \leq \epsilon} \mathcal{L}(h_\theta(\mathbf{x} + \boldsymbol{\delta}), y) \tag{16}$$

Since exactly solving the inner maximization is often not possible, one of the most common approaches of optimization is to alternately solve (approximately) the inner maximization with the help of adversarial attacks and then partially solve the outer minimization using methods such as mini-batch stochastic gradient descent (SGD). Each mini-batch SGD step would require obtaining the adversarial images $\mathbf{x} + \delta^*$, computed with respect to the latest network weight that is updated after the SGD step.

As we would like to use the smooth classifier as the final classifier for robustness properties, it is natural to choose $h_\theta = g_{T,\theta}$ in adversarial training, which has been demonstrated as an effective approach for ANN (Salman et al., 2019). However, there are two difficulties. First, we can only estimate the classifier $g$ using $\hat{g}$, as given in equation 9, thus forcing us to use $h_\theta = \hat{g}_{T,\theta}$. For the computational efficiency of the adversarial training process, we use $m = 1, \alpha = 1$ to obtain a prediction of the smooth classifier, which corresponds to a single evaluation of the rate-encoded classifier. In constant encoding, performing adversarial training corresponds to setting, $h_\theta = f_{T,\theta}$.

Secondly, to carry out the adversarial attack for inner maximization, we need to compute the gradient of $\hat{g}$ with respect to the input, which, following the chain rule, requires back-propagating through the stochastic node that rate-encodes the input $\mathbf{x}$. We employ the Straight Through Estimator (Hinton, 2012; Bengio et al., 2013), which estimates the gradient of a stochastic node assuming there is an identity function in the backward pass of the node, enabling gradient-based attacks to find adversarial examples.

## 5 EXPERIMENTS

**Competitors:** We compare the performance of the proposed methods with state-of-the-art adversarial training algorithms in SNN. We refer to the models trained with un-perturbed data as CLEAN and when perturbed with Gaussian as GN. We generate adversarial images using FGSM and PGD attacks implemented using back-propagation through time (BPTT) with $\epsilon = \frac{8}{255}$ and refer to the corresponding adversarially models by the attack name followed by the abbreviation of encoding, where (C) and (R) stands for constant and rate, respectively. To identify the attacks separately from the corresponding adversarial training algorithms, they are named in small letters, e.g., the FGSM attack assuming a rate-encoded model is referred to as fgsm(R). Further, the training of adversarial

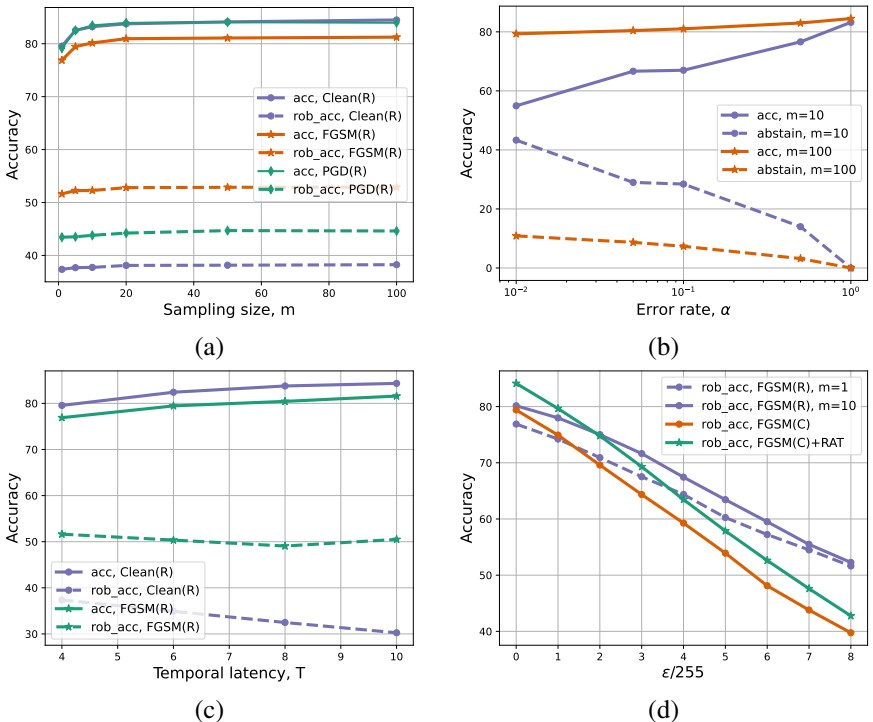

Figure 1: (a) shows the clean and robust accuracy of smooth classifiers improve with m and saturates quickly before m=20. The improvement is more significant for clean accuracy compared to robust accuracy. Fig. (b) shows the effect of error rate $\alpha$ on the prediction. For $m = 100$, the prediction accuracy does not deteriorate at $\alpha = 10^{-2}$, as it helps to avoid the high abstain rate suffered in the case of $m = 10$. Fig. (c) empirically confirms that with larger T, the clean accuracy of rate-encoded classifiers improves, while the robust accuracy drops. Fig. (d) compares robustness of the FGSM(R) model with its competitors, with radius $\epsilon$ of $l_\infty$-norm attacks varying in the range $\left[\frac{0}{255}, \frac{8}{255}\right]$. Prediction of the FGSM(R) model with $m = 10$ offers superior robust accuracy for most cases, while prediction using $m = 1$ beats competitors in the higher range of $\epsilon$.

models, when performed with regularization proposed by Regularized Adversarial Training (Ding et al., 2022), is identified by (RAT). We do not compare with (Kundu et al., 2021) as RAT offers superior adversarial robustness.

**Dataset and Network Architecture:** We use standard static image dataset such as CIFAR-10, CIFAR-100 (Krizhevsky et al., 2009), SVHN (Netzer et al., 2011) and ImageNet-100 (Deng et al., 2009). They are temporally encoded using constant or rate encoding for the SNN classifier to be trained at $T = 4$. We use SEW-Resnet-34 (Fang et al., 2021) architecture for Imagenet-100 and VGGSNN architecture for other datasets. The VGGSNN is based on VGG-11 and has the configuration (64C3-128C3-AP2-256C3-256C3-AP2-512C3-512C3-AP2-512C3-512C3-AP2-FC) (Deng et al., 2022).

**Clean and Robust Accuracy:** The clean accuracy of a model is its accuracy on uncorrupted test images. To evaluate the smooth classifier, we employ Algorithm 1 for the prediction, which returns the output of a conventional rate-encoded network when $m = 1, \alpha = 1$. To demonstrate the smooth classifier, we also make predictions with $m = 10, \alpha = 1$ and study the effect of $\alpha$ on prediction separately, see Fig.1(b). To avoid the scenario where the classifier abstains from prediction due to low statistical confidence, we set $\alpha = 1$, making the accuracies comparable to the constant models. The empirical robust accuracy is the prediction accuracy when input images are corrupted with specific adversarial attacks. With the exception to Fig.1(d) where we study the effect of attack radius, the $l_\infty$-norm radius is set to $\epsilon = \frac{8}{255}$.

## 5.1 COMPARISON OF EMPIRICAL ROBUST ACCURACY

Table 1 compares the constant encoded network under different training algorithms with the rate-encoded counterpart. We highlight the minimum accuracy against all attacks for each training method, reported column-wise. The minimum robust accuracy represents the empirical robust-

| T=4 | CIFAR-10, Constant Encoding | | | | CIFAR-100, Constant Encoding | | | |
|---|---|---|---|---|---|---|---|---|
| Attack | CLEAN | GN | FGSM(C) | PGD(C) | CLEAN | GN | FGSM(C) | PGD(C) |
| clean | 92.15 | 91.7 | 79.4 | 79.15 | 72.01 | 70.19 | 54.31 | 53.38 |
| gn | 90.62 | 91.25 | 78.84 | 78.28 | 66.46 | 69.64 | 53.87 | 53.02 |
| fgsm(C) | 10.68 | 15.95 | 48.86 | 48.56 | 3.19 | 5.35 | 26.05 | 26.49 |
| pgd(C) | **0.1** | **1.24** | **39.75** | **41.65** | **0.04** | **0.45** | **21.12** | **22.61** |
| fgsm(R) | 81.01 | 85.13 | 77.33 | 75.69 | 45.4 | 59.23 | 49.63 | 50.81 |
| pgd(R) | 85.14 | 85.07 | 77.42 | 75.29 | 55.82 | 59.73 | 49.57 | 50.69 |

| T=4 | CIFAR-10, Constant Encoding, RAT | | | | CIFAR-100, Constant Encoding, RAT | | | |
|---|---|---|---|---|---|---|---|---|
| Attack | CLEAN | GN | FGSM(C) | PGD(C) | CLEAN | GN | FGSM(C) | PGD(C) |
| clean | 91.29 | 90.86 | 84.13 | 83.33 | 70.85 | 68.87 | 58.1 | 58.05 |
| gn | 88.71 | 90.72 | 83.09 | 82.95 | 66.05 | 69.82 | 57.93 | 57.34 |
| fgsm(C) | 25.09 | 29.42 | 54.18 | 53.50 | 10.39 | 13.12 | 35.62 | 34.36 |
| pgd(C) | **0.49** | **3** | **42.76** | **44.44** | **0.34** | **1.71** | **27.1** | **29.89** |
| fgsm(R) | 84.62 | 87.55 | 81.68 | 80.90 | 53.15 | 64.34 | 56.31 | 55.08 |
| pgd(R) | 87.41 | 87.96 | 81.96 | 80.58 | 62.88 | 64.93 | 55.85 | 54.52 |

| T=4, m=1 | CIFAR-10, Rate Encoding | | | | CIFAR-100, Rate Encoding | | | |
|---|---|---|---|---|---|---|---|---|
| Attack | CLEAN | GN | FGSM(R) | PGD(R) | CLEAN | GN | FGSM(R) | PGD(R) |
| clean | 79.55 | 79.36 | 76.89 | 76.36 | 50.9 | 50.89 | 45.85 | 46.98 |
| gn | 78.62 | 79.07 | 76.39 | 76.77 | 50.43 | 50.77 | 46.4 | 46.05 |
| fgsm(C) | 75.57 | 75.34 | 72.03 | 73.99 | 48.36 | 49.12 | 45.16 | 45.15 |
| pgd(C) | 76.23 | 76.06 | 73.03 | 78.87 | 47.94 | 48.95 | 45.74 | 45.38 |
| fgsm(R) | 43.69 | 43.31 | 55.05 | 55.27 | 25.75 | 24.84 | 31.64 | 32.47 |
| pgd(R) | **37.37** | **37.4** | **51.63** | **51.59** | **21.98** | **20.72** | **28.56** | **29.99** |

| T=4, m=10 | CIFAR-10, Rate Encoding | | | | CIFAR-100, Rate Encoding | | | |
|---|---|---|---|---|---|---|---|---|
| Attack | CLEAN | GN | FGSM(R) | PGD(R) | CLEAN | GN | FGSM(R) | PGD(R) |
| clean | 83.22 | 83.5 | 80.15 | 80.54 | 55.27 | 55.83 | 49.58 | 50.51 |
| gn | 83.54 | 83.41 | 80.33 | 80.41 | 54.82 | 55.2 | 49.84 | 50.15 |
| fgsm(C) | 80.29 | 79.94 | 76.41 | 78.22 | 52.76 | 53.6 | 49.19 | 48.3 |
| pgd(C) | 80.6 | 80.05 | 76.73 | 78.59 | 52.54 | 53.73 | 49.07 | 48.8 |
| fgsm(R) | 44.93 | 44.89 | 57.89 | 57.85 | 27.61 | 26.32 | 33.46 | 33.88 |
| pgd(R) | **37.74** | **37.58** | **52.27** | **53.08** | **22.79** | **21.5** | **30.03** | **31.05** |

Table 1: Experiments on CIFAR-10 and CIFAR-100 datasets demonstrate that under the strongest attacks, rate encoded classifiers offer superior robustness compared to the constant encoded counterpart. The columns stands for different training (adversarial) procedure, while the rows stands for different adversarial attacks.

ness of a model when subjected to arbitrary attack. It is found that for constant-encoded models, pgd(C) attack gives minimum accuracy, while for the rate-encoded models, pgd(R) is the strongest. It can be observed that RAT improves the robustness of the constant encoded models. However, rate-encoded models significantly improve the robust accuracy compared to RAT. For example, in CIFAR-10, CLEAN(R) is 37% more robust than CLEAN(C), while under adversarial training, FGSM(R), PGD(R) improve the minimum robust accuracy by 9% and 18%, respectively, compared to RAT. For CIFAR-100, the improvement in robust accuracy is 22% for CLEAN(R), 3% for FGSM(R), and 13% for PGD(R), considering predictions using $m = 10$.

A common criticism of the rate-encoded model is that it offers lower accuracy on clean images than constant encoding. Fig.1(a) shows on CIFAR-10 that the prediction of rate-encoded models can significantly improve with the better estimation of the smooth model at the test time. The same observation can be made from Table 1 where shifting from m=1 to 10 improves the clean accuracy by 4-5 % across all the rate-encoded models. A further defense for rate-encoded models comes from Fig.1(d), where we vary radius of the $l_\infty$-attack between $[\frac{0}{255}, \frac{8}{255}]$, where $\epsilon = 0$, corresponds to clean images. It can be observed that constant encoded models quickly lose accuracy even for small perturbations, showing the relevance of rate-encoded models when there is a possibility of attack. Fig.1(c) studies the effect of latency on robustness. It separately trains the CLEAN(R) and FGSM(R) models at various $T \in \{4, 6, 8, 10\}$ and measures their clean and robust accuracy (against pgd(R)). The results show that although clean accuracy improves with latency, the robust accuracy

| ImageNet-100, T=4 | clean | gn | fgsm(C) | pgd(C) | fgsm(R) | pgd(R) |
|---|---|---|---|---|---|---|
| CLEAN(C) | 72.02 | 71.78 | 4.98 | **0.02** | 51.18 | 61.1 |
| CLEAN(C) +RAT | 65.14 | 63.66 | 7.48 | **0.06** | 51.1 | 57.22 |
| CLEAN(R), m=1 | 62.16 | 62 | 52.3 | 55.44 | 19.88 | **12.06** |
| CLEAN(R), m=10 | 64.18 | 64.1 | 58.42 | 60.24 | 28.04 | **19.86** |

| SVHN, T=4 | clean | gn | fgsm(C) | pgd(C) | fgsm(R) | pgd(R) |
|---|---|---|---|---|---|---|
| CLEAN(C) | 95.36 | 94.71 | 26.6 | **3.11** | 65.61 | 70.56 |
| CLEAN(C) +RAT | 96.17 | 95.5 | 40.43 | **4.03** | 80.33 | 78.44 |
| CLEAN(R), m=1 | 86.09 | 85.67 | 77.49 | 77.82 | 43.68 | **37.44** |
| CLEAN(R), m=10 | 91.7 | 91.44 | 84.51 | 84.87 | 46.58 | **38.79** |

Table 2: Experiments on ImageNet-100 and SVHN show that rate-encoded models can offer reasonable clean accuracy and significantly higher robust accuracy against the strongest attacks.

| T=4 | CIFAR-10, Rate Encoding | | | | | CIFAR-100, Rate Encoding | | | | |
|---|---|---|---|---|---|---|---|---|---|---|
| $r * 108 =$ | 0.1 | 0.2 | 0.3 | 0.4 | 0.45 | 0.1 | 0.2 | 0.3 | 0.4 | 0.45 |
| CLEAN(R) | 72.48 | 67.62 | 60.29 | 49.71 | 37.01 | 41.8 | 37.62 | 32.02 | 25.36 | 18.49 |
| PGD(R) | 73.23 | 68.5 | 61.69 | 51.58 | 38.78 | 41.48 | 37.31 | 32.15 | 25.65 | 18.68 |
| T=4 | ImageNet-100, Rate Encoding | | | | | SVHN, Rate Encoding | | | | |
| CLEAN(R) | 51.38 | 46.7 | 41.04 | 33.54 | 24.52 | 81.67 | 76.85 | 69.15 | 56.67 | 41.18 |

Table 3: presents the certified test accuracy for rate-encoded SNN models at different $l_1$-norm radii for various static datasets.

drops, confirming our theoretical results. In the appendix, we provide full results for $T = 8$ to further confirm the same.

**Results on other Datasets:** We further compare the robustness of constant vs. rate-encoding on larger datasets. Often, for large datasets such as ImageNet, adversarial training can be computationally challenging due to the size of the dataset, which gets multiplied by the temporal dimension of SNN. However, rate-encoding encoded classifiers trained with clean images offer robustness properties and are computationally easier to train. Table2 provides empirical robustness results for Imagenet-100 and SVHN dataset, where CLEAN(R) obtains significantly higher robust accuracy than other feasible methods.

## 5.2 PROVABLE ROBUSTNESS VIA CERTIFIED ACCURACY

We provide the empirical certified test accuracy of the rate-encoded smooth classifier for perturbation bounded under $l_1$-norm radius. As given in Table 3, the radius varies between $\left[\frac{0.1}{L(T)}, \frac{0.5}{L(T)}\right]$, with $L(T) = T3^{T-1}$ representing the Lipschitzness of the smooth classifier as in Lemma 4. We use the certify function with parameters $m_0 = 10$, $m = 100$, and $\alpha = 0.01$ to find the certified accuracy of the rate-encoded models. The certified radii obtained are not entirely vacuous. For example, at $T = 4$, we have $L(T) = 108$, so that we can obtain a certified accuracy at the radius $\frac{0.45}{108}$ which is larger than $\frac{1}{255}$. Assuming 8-bit pixels allows a single pixel in the image to be changed by 1. There also remains scope to obtain the certificate at lower latencies, as highlighted in the section 3.2. Fig. 2 shows a comparison between certified accuracy and robust accuracy under $l_1$ attack.

## 6 DISCUSSIONS

The present work provides a theoretical grounding for the empirical observations of the adversarial robustness of the rate-encoded classifiers. Consequently, we improve the classifier's prediction at test time by better estimating the corresponding smooth classifier. Further, we improve the empirical robustness of the rate-encoded classifiers by adversarial training, which uses a novel implementation of gradient-based attacks. The future scope for research remains open to reduce the gap between empirical and theoretical results on the shrinkage of the certified radius with latency and other norm-based attacks.

ACKNOWLEDGEMENT

This work is part of the research project "ENERGY-BASED PROBING FOR SPIKING NEURAL NETWORKS" performed at Mohamed bin Zayed University of Artificial Intelligence (MBZUAI), in collaboration with Technology Innovation Institute (TII) (Contract No. TII/ARRC/2073/2021). We thank Velibor Bojkovic for proofreading the theorems.

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

## A APPENDIX

**Lemma 3.** *For all $y$, $g(\mathbf{x})_y$ is 1-Lipschitz under $\|\cdot\|_1$-norm.*

*Proof.*

$$
\begin{aligned}
g(\mathbf{x})_y = \mathbb{E}_{z_i \sim Ber(x_i)}[f(\langle z_1, z_2, \cdots, z_d \rangle)_y] &= \sum_{z_i^{(j)} \in \{0,1\}} f(\langle z_1^{(j)}, z_2^{(j)}, \cdots, z_d^{(j)} \rangle)_y \prod_{i=1}^{d} \mathbb{P}(z_i^{(j)}|x_i) \\
&= \sum_{z_i^{(j)} \in \{0,1\}} f(\langle z_1^{(j)}, z_2^{(j)}, \cdots, z_d^{(j)} \rangle)_y \prod_{i=1}^{d} (x_i)^{z_i^{(j)}} (1-x_i)^{1-z_i^{(j)}} \\
&= \sum_{i \neq 1, z_i^{(j)} \in \{0,1\}} f(\langle 1, z_2^{(j)}, \cdots, z_d^{(j)} \rangle)_y \, x_1 \prod_{i \neq 1} (x_i)^{z_i^{(j)}} (1-x_i)^{1-z_i^{(j)}} \\
&\qquad + \sum_{i \neq 1, z_i^{(j)} \in \{0,1\}} f(\langle 0, z_2^{(j)}, \cdots, z_d^{(j)} \rangle)_y \, (1-x_1) \prod_{i \neq 1} (x_i)^{z_i^{(j)}} (1-x_i)^{1-z_i^{(j)}}
\end{aligned}
$$

Let us now take partial derivative of $g(\cdot)_y$ w.r.t. $x_1$, which can be generalized to other co-ordinates in a similar fashion:

$$
\begin{aligned}
\left| \frac{\partial g_y}{\partial x_1} \right| &= \sum_{i \neq 1, \, z_i^{(j)} \in \{0,1\}} \left| \left( f(\langle 1, z_2^{(j)}, \cdots, z_d^{(j)} \rangle)_y - f(\langle 0, z_2^{(j)}, \cdots, z_d^{(j)} \rangle)_y \right) \right| \prod_{i \neq 1} (x_i)^{z_i^{(j)}} (1-x_i)^{1-z_i^{(j)}} \\
&\leq \sum_{i \neq 1, \, z_i^{(j)} \in \{0,1\}} \prod_{i \neq 1} (x_i)^{z_i^{(j)}} (1-x_i)^{1-z_i^{(j)}} \qquad \text{[diff. of probabilities is less than 1]} \\
&= 1 \qquad \text{[using marginal probability]}
\end{aligned}
$$

Thus, we have, $\|\nabla g(\cdot)_y\|_\infty \leq 1$ which following the Definition 1, ensures $g(\cdot)_y$ is $1-$Lipschitz under $\|\cdot\|_1$-norm. $\qquad\square$

**Lemma 4.** *For all $y$, $g_T(\mathbf{x})_y$ is $T3^{T-1}$-Lipschitz under $\|\cdot\|_1$-norm.*

*Proof.*

$$
\begin{aligned}
g_T(\mathbf{x}) = \mathbb{E}_{\mathbf{z}_i \sim Ber^T(x_i)}[\tilde{f}(\langle \mathbf{z}_1, \mathbf{z}_2, \cdots, \mathbf{z}_d \rangle)] & \\
&= \sum_{\mathbf{z}_i \in \{0,1\}^T} \tilde{f}(\langle \mathbf{z}_1, \mathbf{z}_2, \cdots, \mathbf{z}_d \rangle) \prod_{i=1}^{d} \prod_{t=1}^{T} \mathbb{P}(z_{i,t}|x_i) \\
&= \sum_{\mathbf{z}_i \in \{0,1\}^T} \tilde{f}(\langle \langle z_{1,1}, \cdots, z_{1,T} \rangle, \mathbf{z}_2, \cdots \mathbf{z}_d \rangle) \prod_{t=1}^{T} x_1^{z_{1,t}} (1-x_1)^{1-z_{1,t}} \prod_{i \neq 1} \prod_{t=1}^{T} \mathbb{P}(z_{i,t}|x_i)
\end{aligned}
$$

Now, let us denote, $c_k(\mathbf{z}_2, \cdots, \mathbf{z}_d) = \sum_{\|\mathbf{z}_1\|_1 = k} \tilde{f}(\langle \langle \cdot \rangle, \mathbf{z}_2, \cdots, \mathbf{z}_d \rangle)_y$, the sum of $\tilde{f}$ for different possible values that $\mathbf{z}_1$ can assume, with the restriction that there are $k$ 1's out of $T$ binary values. Thus,

$$0 \le c_0(\mathbf{z}_2, \cdots, \mathbf{z}_d) := \tilde{f}(\langle \langle 0, 0, \cdots 0 \rangle, \mathbf{z}_2, \cdots, \mathbf{z}_d \rangle)_y \le \binom{T}{0}$$

$$0 \le c_1(\mathbf{z}_2, \cdots, \mathbf{z}_d) := \tilde{f}(\langle \langle 1, 0, \cdots 0 \rangle, \mathbf{z}_2, \cdots, \mathbf{z}_d \rangle)_y + \cdots + \tilde{f}(\langle \langle 0, 0, \cdots, 1 \rangle, \mathbf{z}_2, \cdots, \mathbf{z}_d \rangle)_y \le \binom{T}{1}$$

$$\vdots$$

$$0 \le c_T(\mathbf{z}_2, \cdots, \mathbf{z}_d) := \tilde{f}(\langle \langle 1, 1, \cdots, 1 \rangle, \mathbf{z}_2, \cdots, \mathbf{z}_d \rangle)_y \le \binom{T}{T}$$

and, $P(x_2, \cdots, x_d) = \prod_{i \ne 1} \prod_{t=1}^{T} \mathbb{P}(z_{i,t}|x_i)$

So that, we may write after expanding the expectation over the values that variable $\mathbf{z}_1$ may take:

$$g_T(\mathbf{x}) = \sum_{\mathbf{z}_i \in \{0,1\}^T} \tilde{f}(\langle \langle z_{1,1}, \cdots, z_{1,T} \rangle, \mathbf{z}_2, \cdots \mathbf{z}_d \rangle) \prod_{t=1}^{T} x_1^{z_{1,t}} (1-x_1)^{1-z_{1,t}} P(x_2, \cdots, x_d)$$

$$= \sum_{i \ne 1, \mathbf{z}_i \in \{0,1\}^T} \sum_{k=0}^{T} c_k x_1^k (1-x_1)^{T-k} P(x_2, \cdots, x_d)$$

Let us denote $h : [0,1] \to \mathbb{R}$ as the function :

$$h_T(x) = \sum_{k=0}^{T} c_k x^k (1-x)^{T-k} = \sum_{k=0}^{T} c_k x^k \sum_{j=0}^{T-k} \binom{T-k}{j} (-x)^j$$

$$= \sum_{k=0}^{T} \sum_{j=0}^{T-k} c_k \binom{T-k}{j} x^{k+j} (-1)^j$$

$$= \sum_{l=0}^{T} \sum_{m=0}^{l} c_m \binom{T-m}{l-m} x^l (-1)^{l-m} \quad \text{[after re-indexing]}$$

$$\frac{dh_T(x)}{dx} = \sum_{l=1}^{T} l x^{l-1} \sum_{m=0}^{l} c_m \binom{T-m}{l-m} (-1)^{l-m}$$

$$= \sum_{l=1}^{T} l x^{l-1} \sum_{j=0}^{l} c_{l-j} \binom{T-l+j}{j} (-1)^j \quad \text{[setting, j=l-m]}$$

$$\le \sum_{l=1}^{T} l x^{l-1} \sum_{j=0}^{l/2} \binom{T}{l-2j} \binom{T-l+2j}{2j} \quad as, 0 \le c_j \le \binom{T}{j}$$

$$= \sum_{l=1}^{T} l x^{l-1} \sum_{j=0}^{l/2} \frac{T!}{(l-2j)!(T-l)!(2j)!}$$

$$= \sum_{l=1}^{T} l x^{l-1} \binom{T}{l} \sum_{j=0}^{l/2} \frac{l!}{(l-2j)!(2j)!}$$

$$= \sum_{l=1}^{T} l x^{l-1} \binom{T}{l} 2^{l-1} = T \sum_{l=1}^{T} \binom{T-1}{l-1} (2x)^{l-1} = T(1+2x)^{T-1} \le T 3^{T-1}$$

Taking partial derivative of $g$ w.r.t. $x_1$,

$$\left|\frac{\partial g_T(\cdot)_y}{\partial x_1}\right| = \sum_{i \neq 1, \mathbf{z}_i \in \{0,1\}^T} |h'_T(x_1)| P(x_2, \cdots, x_d)$$

$$\leq T3^{T-1} \sum_{i \neq 1, \mathbf{z}_i \in \{0,1\}^T} P(x_2, \cdots, x_d)$$

$$= T3^{T-1}$$

Thus, we have, $\|\nabla g\|_\infty \leq T3^{T-1}$ which following the Definition 1, ensures $g$ is $T3^{T-1}-$Lipschitz under norm $\|\cdot\|_1$.

$\square$

## B   ADDITIONAL EXPERIMENTS

**Comparison at higher latency** Table 4 compares the constant encoded SNN under different training algorithms with their rate encoded counterpart at $T = 8$. Similar to Table1 with $T = 4$, adversarially trained rate-encoding demonstrates superior accuracy compared to their constant encoded counterparts. Also, one can observe that for most cases in rate-encoded models, the clean accuracy has improved when compared to $T = 4$, while the robust accuracy has dropped. Our theoretical findings for rate-encoded models also confirm that rate-encoded classifiers' robustness decreases with higher latency.

| T=8 | CIFAR-10, Constant Encoding | | | | CIFAR-100, Constant Encoding | | | |
|---|---|---|---|---|---|---|---|---|
| Attack | CLEAN | GN | FGSM(C) | PGD(C) | CLEAN | GN | FGSM(C) | PGD(C) |
| clean | 90.29 | 92.28 | 81.71 | 82.69 | 73 | 71.2 | 55.21 | 56.96 |
| gn | 88.93 | 91.85 | 81.26 | 82.56 | 68.67 | 71 | 53.04 | 56.47 |
| fgsm(C) | 8.45 | 18.04 | 48.79 | 46.33 | 4.17 | 6.47 | 32.39 | 25.99 |
| pgd(C) | **0.07** | **1.11** | **39.41** | **41.04** | **0.1** | **0.67** | **17.37** | **23.36** |
| fgsm(R) | 71.03 | 84.77 | 79.17 | 80.43 | 49.36 | 62.03 | 52.06 | 53.39 |
| pgd(R) | 75.54 | 84.49 | 79.48 | 81.35 | 61.36 | 62.84 | 51.01 | 54.55 |

| T=8,m=1 | CIFAR-10, Rate Encoding | | | | CIFAR-100, Rate Encoding | | | |
|---|---|---|---|---|---|---|---|---|
| Attack | CLEAN | GN | FGSM(R) | PGD(R) | CLEAN | GN | FGSM(R) | PGD(R) |
| clean | 83.77 | 83.57 | 80.43 | 80.64 | 54.26 | 48.73 | 48.78 | 50.85 |
| gn | 76.94 | 82.86 | 80.03 | 79.76 | 53.34 | 48.25 | 48.58 | 50.41 |
| fgsm(C) | 70.49 | 78.85 | 76.63 | 77.05 | 51.36 | 42.6 | 47.87 | 48.34 |
| pgd(C) | 71.76 | 80.11 | 77.8 | 77.86 | 50.9 | 44.1 | 47.46 | 49.22 |
| fgsm(R) | 36.05 | 41.01 | 54.42 | 52.02 | 24.95 | 17.74 | 33.51 | 28.87 |
| pgd(R) | **32.48** | **31.82** | **49.06** | **51.29** | **21.23** | **14.13** | **33.41** | **28.31** |

Table 4: Experiments on CIFAR-10 and CIFAR-100 datasets, with $T = 8$, demonstrate that under the strongest attacks, rate encoded classifiers offer superior robustness compared to the constant encoded counterpart. The columns stands for different training (adversarial) procedure, while the rows stands for different adversarial attacks.

**Comparison of empirical robust accuracy:** We further compare the test fgsm/pgd attacks implemented with different back-propagation techniques such as, Back Propagation Through Time (BPTT) (Wu et al., 2018), Backward Pass Through Rate (BPTR) (Ding et al., 2022), Rate Gradient Attack (RGA) (Bu et al., 2023), and report the corresponding accuracies in the respective order. Table5 shows the experiments conducted on CIFAR-10, where BPTT produces the strongest attacks.

**Comparison under Gaussian noise attack** The results in Table 1 with gn attack uses Gaussian perturbation: $x + \delta$, where $\delta \sim N(0, \sigma^2)$, with $\sigma = 8/255 \approx 0.031$. The superior results obtained there for the constant encoded models against Gaussian noise attacks can quickly break down as we increase the strength of the attacks, as computed in Table6. For example, at $\sigma = 0.1$, CLEAN(R) with m=10, provides 21.77% higher accuracy than CLEAN(C)+RAT.

| T=4 | CIFAR-10, Constant Encoding | | | |
|---|---|---|---|---|
| Attack | CLEAN | GN | FGSM(C) | PGD(C) |
| clean | 92.15 | 91.7 | 79.4 | 79.15 |
| gn | 90.62 | 91.25 | 78.84 | 78.28 |
| fgsm(C) | 10.68 \| 16.08\|10.68 | 15.95 \| 22.40 \| 15.53 | 48.86 \| 55.79 \| 52.73 | 48.56 \| 55.55 \| 52.12 |
| pgd(C) | **0.1** \| 1.71 \| 0.47 | **1.24** \| 6.64 \| 2.76 | **39.75** \| 49.70 \| 47.62 | **41.65** \| 49.57 \| 48.30 |
| fgsm(R) | 81.01 \| 84.36 \| 80.18 | 85.13 \| 88.39 \| 83.10 | 77.33 \| 78.08 \| 76.78 | 75.69 \| 76.93 \| 74.90 |
| pgd(R) | 85.14 \| 88.84 \| 82.49 | 85.07 \| 88.73 \| 82.48 | 77.42 \| 78.40 \| 77.02 | 75.29 \| 76.46 \| 75.13 |
| T=4 | CIFAR-10, Constant Encoding, RAT | | | |
| Attack | CLEAN | GN | FGSM(C) | PGD(C) |
| clean | 91.29 | 90.86 | 84.13 | 83.33 |
| gn | 88.71 | 90.72 | 83.09 | 82.95 |
| fgsm(C) | 25.09 \| 25.90 \| 15.40 | 29.42 \| 34.45 \| 20.74 | 54.18 \| 64.46 \| 55.86 | 53.50 \| 63.97 \| 56.15 |
| pgd(C) | 0.49 \| 5.97 \| **0.45** | **3** \| 15.93 \| 3.23 | **42.76** \| 58.92 \| 48.66 | **44.44** \| 58.78 \| 50.05 |
| fgsm(R) | 84.62 \| 84.06 \| 82.94 | 87.55 \| 88.59 \| 87.11 | 81.68 \| 82.56 \| 81.45 | 80.90 \| 82.53 \| 0.63 |
| pgd(R) | 87.41 \| 88.29 \| 85.75 | 87.96 \| 89.15 \| 87.41 | 81.96 \| 82.95 \| 81.60 | 80.58 \| 82.52 \| 80.33 |
| T=4, m=1 | CIFAR-10, Rate Encoding | | | |
| Attack | CLEAN | GN | FGSM(R) | PGD(R) |
| clean | 79.55 | 79.36 | 76.89 | 76.36 |
| gn | 78.62 | 79.07 | 76.39 | 76.77 |
| fgsm(C) | 75.57 \| 78.01 \| 75.62 | 75.34 \| 77.75 \| 75.36 | 72.03 \| 73.01 \| 73.00 | 73.99 \| 75.81 \| 74.50 |
| pgd(C) | 76.23 \| 78.73 \| 76.19 | 76.06 \| 78.74 \| 76.30 | 73.03 \| 74.76 \| 73.35 | 74.87 \| 76.62 \| 75.15 |
| fgsm(R) | 43.69 \| 55.97 \| 46.02 | 43.31 \| 54.90 \| 45.30 | 55.05 \| 62.49 \| 56.06 | 55.27 \| 63.42 \| 55.77 |
| pgd(R) | **37.37** \| 51.71 \| 39.48 | **37.4** \| 50.79 \| 40.0 | **51.63** \| 59.92 \| 52.12 | **51.59** \| 61.21 \| 51.99 |
| T=4, m=10 | CIFAR-10, Rate Encoding | | | |
| Attack | CLEAN | GN | FGSM(R) | PGD(R) |
| clean | 83.22 | 83.5 | 80.15 | 80.54 |
| gn | 83.54 | 83.41 | 80.33 | 80.41 |
| fgsm(C) | 80.29 \| 82.65 \| 80.05 | 79.94 \| 82.04 \| 80.37 | 76.41 \| 77.97 \| 76.82 | 78.22 \| 79.88 \| 78.6 |
| pgd(C) | 80.6 \| 82.93 \| 80.51 | 80.05 \| 82.73 \| 80.27 | 76.73 \| 78.87 \| 76.76 | 78.59 \| 80.15 \| 78.69 |
| fgsm(R) | 44.93 \| 59.58 \| 47.64 | 44.89 \| 57.75 \| 47.54 | 57.89 \| 65.94 \| 58.63 | 57.85 \| 66.87 \| 59.03 |
| pgd(R) | **37.74** \| 53.68 \| 40.97 | **37.58** \| 52.2 \| 40.46 | **52.27** \| 62.07 \| 53.77 | **53.08** \| 63.93 \| 54.19 |

Table 5: We compare the test fgsm/pgd attacks implemented with different back-propagation techniques such as BPTT, BPTR, and RGA and report the corresponding accuracies in the respective order. The results show that, most often, BPTT produces the most potent attacks.

| $\sigma$ | 0.05 | 0.1 | 0.2 | 0.3 | 0.4 | 0.5 |
|---|---|---|---|---|---|---|
| CLEAN(C) | 86.53 | 59.58 | 22.8 | 14.63 | 13.14 | 11.73 |
| CLEAN(C)+RAT | 84.3 | 60.3 | 26.01 | 16.01 | 12.66 | 11.26 |
| CLEAN(R), m=1 | 78.38 | 77.74 | 72.58 | 61.77 | 48.38 | 35.13 |
| CLEAN(R), m=10 | 83.13 | 82.07 | 77.65 | 66.26 | 51.14 | 36.66 |

Table 6: Under stronger Gaussian noise rate-encdoed SNNs show superior robustness.

| | Constant Encoding | | | | Rate encoding | | | |
|---|---|---|---|---|---|---|---|---|
| **Attack** | **T**=4 | 32 | 64 | 128 | 4 | 32 | 64 | 128 |
| clean | 92.58 | 95.47 | 95.47 | 95.51 | 11.83 | 24.12 | 41.5 | 63.11 |
| gn | 85.84 | 89.1 | 88.98 | 88.97 | 11.54 | 22.8 | 38.35 | 57.4 |
| fgsm(C) | 21.02 | 14.25 | 15.85 | 17.56 | 11.14 | 17.43 | 23.64 | 28.8 |
| pgd(C) | 0.2 | 0.03 | 0.02 | 0.03 | 11.24 | 17.56 | 23.54 | 26.2 |
| fgsm(R) | 73.59 | 64.54 | 37.34 | 14.7 | 11.49 | 16.36 | 18 | 15.76 |
| pgd(R) | 87.46 | 82.53 | 58.36 | 11.9 | 11.46 | 17.54 | 18.94 | 13.92 |

Table 7: Converted SNN model trained on CIFAR10 dataset

Table 8: Training hyper-parameters

| | CIFAR-10/100 | ImageNet-100 | SVHN |
|---|---|---|---|
| Number epochs | 200 | 200 | 200 |
| Mini batch size | 64 | 64 | 64 |
| T | 4,8 | 4 | 4 |
| LIF: $\beta$ | 0.5 | 1 | 0.5 |
| LIF: $u_0$ | 0 | 0 | 0 |
| LIF: $u_{th}$ | 1 | 1 | 1 |
| Learning Rate | 0.1 | 0.1 | 0.1 |
| FGSM/PGD/GN: $\epsilon$ | 8/255 | 8/255 | 8/255 |
| PGD (train): $\eta$ | 2/255 | na | na |
| PGD (train) Iteration | 4 | na | na |
| PGD (test): $\eta$ | 2.55/255 | 2.55/255 | 2.55/255 |
| PGD (test) Iteration | 7 | 7 | 7 |

Optimizer: SGD with momentum: 0.9, weight decay: $5 \times 10^{-4}$, Rate Scheduler: cosine annealing

**Comparison under ANN-SNN converted models:** The theory of randomized smoothing holds irrespective of the base classifier that we choose. That is, whether we use the base classifier $f$ (in eqn. 6) from an adversarially trained model or a converted model, the corresponding smooth classifier g, will be robust against adversarial perturbation, i.e., $g(x + \delta) = g(x)$. However, if the base classifier is unfit, the prediction $g(x)$ can be incorrect, leading to poor clean and robust accuracy.

We conducted experiments on VGG-16 models obtained using ANN-to-SNN conversion(Bu et al., 2021) on the CIFAR-10 dataset. Under rate-encoding, converted SNNs do not offer clean accuracy at smaller latency T, which should be considered as the limitation of the base classifier trained to work with direct inputs without the Bernoulli noise. The same happens to constant encoded SNNs trained on clean images if we add significant Gaussian noise to the input (see Table6).

It is interesting to note that, (i) while the clean accuracy of rate encoding is lower than that of directly trained SNNs, the robust accuracy (minimum accuracy across among the attacks) of rate encoding surpasses that of SNNs with constant encoding. (ii) The clean accuracy of rate encoding keeps improving larger T, but the robust accuracy initially increases (due to better prediction), but eventually drops, possibly due to the effect of T as found in our theory. (T=4: 11.49, T=32:16.36, T=64: 18, T=128: 15.76). A similar observation was made in directly trained SNNs, as given in Table 4, as we evaluated the direct trained models at T=4 vs T=8. (iii) A similar drop in robust accuracy for constant encoded models may hint at rate-encoded nature of spikes in general SNN layers, as reported in (Bu et al., 2023).

**Training hyper-parameters and time** Table 8 reports the training hyper-parameters used across the four datasets. Additionally, Table 9 reports time required by each epoch of various adversarial training methods on a single NVIDIA RTX A6000 GPU. For example, the column FGSM(C/R) report that both constant and rate encoded adversarial training takes same amount of time for a single epoch.

**Comparison with $l_1$ adversarial perturbations:** The certified accuracy is further compared with robust accuracy obtained with respect to the pgd attack within an $l_1$ adversarial budget. Projection

Table 9: Training time per epoch (in seconds)

|  | CLEAN (C/R) | GN (C/R) | FGSM(C/R) | PGD(C/R) |
|---|---|---|---|---|
| CIFAR-10 | 65 | 66 | 106 | 150 |
| CIFAR-100 | 48 | 48 | 72 | 251 |
| Imagenet-100 | 704 | na | na | na |
| SVHN | 95 | na | na | na |

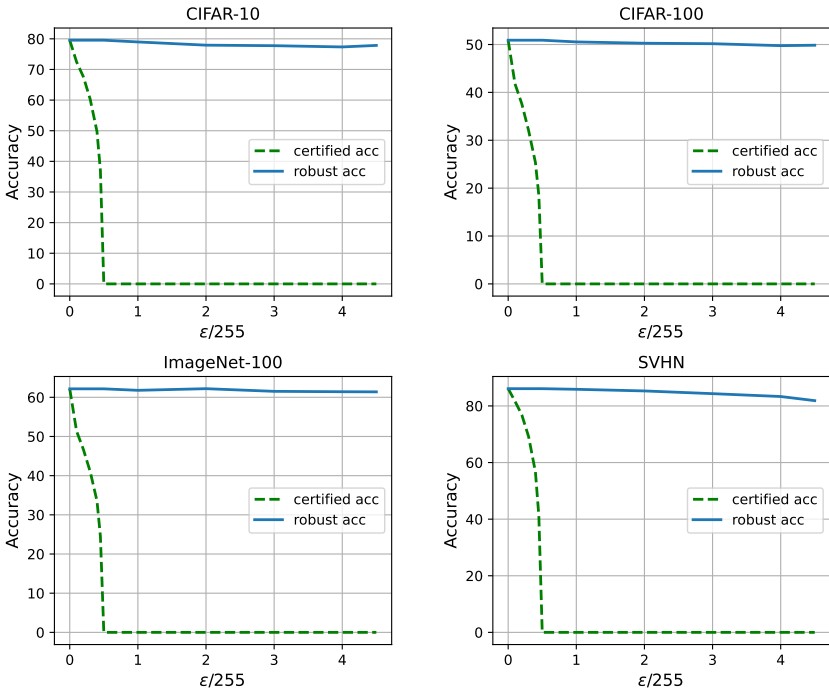

Figure 2: We compare the empirical robust accuracy of a CLEAN(R) model under the PGD attack with its certified accuracy across various $l_1$ radius values and for different datasets. The corresponding gap highlights the scope for improvement in theoretical/experimental results.

into $l_1$ ball is implemented using the code from (Croce & Hein, 2021). Figure 2 compares the results on different datasets.

