# OpenReview forum: "Certified Adversarial Robustness for Rate Encoded Spiking Neural Networks"
_ICLR.cc/2024/Conference — ICLR 2024 poster_

### Official Review · Reviewer_FaRA · 2023-10-29

**Soundness:** 2 fair
**Presentation:** 2 fair
**Contribution:** 2 fair
**Rating:** 8
**Confidence:** 3

**Summary:**

This paper pointed out that rate-encoded SNNs have superior robustness compared to constant-encoded SNNs, and then proposed an adversarial training method based on rate encoding.

**Strengths:**

1. The author's relevant mathematical derivation about the perturbation degree $\boldsymbol{\delta}$ is quite convincing, especially the connection between $\boldsymbol{\delta}$ and $T$ established in Theorem 5.

**Weaknesses:**

1. As shown in Tab.1, compared to constant-encoded SNN, rate-encoded SNN seems to have a significant loss in its clean accuracy (CIFAR10: 91.29% v.s. 83.22%, CIFAR-100: 70.85% v.s. 55.27%). I think this will significantly hinder the practical application of rate-encoded SNN.

2. In Tab.1, the author's comparative perspective is that the robustness of constant-encoded SNN under PGD (C) is weaker than that of rate-encoded SNN under PGD (R), so rate-encoded SNN is better. However, I noticed that the robustness of constant-encoded SNN is better than that of rate-encoded SNN under GN attacks (CIFAR10: 90.62% v.s. 78.62%, CIFAR-100: 66.46% v.s. 50.43%), and at the same time, constant-encoded SNN maintains good defense against FGSM (R) and PGD (R). **Therefore, it seems that the so-called superior robustness of rate-encoded SNN does not have a good generalization effect, and it cannot maintain good defense ability under different attacks, which is an obvious limitations.**

**Questions:**

See Weakness Section.

---

> ### Author Response · Authors · 2023-11-20
>
> **As shown in Tab.1, compared to constant-encoded SNN, rate-encoded SNN seems to have a significant loss in its clean accuracy. I think this will significantly hinder the practical application of rate-encoded SNN.**
>
> Ans: Indeed, it is well known that rate-encoding has the limitation of lower clean accuracy, especially for low latency inference. However, we would like to emphasize that the proposed connection to randomized smoothing reveals that by better approximating the smooth classifier (i.e., using larger m), we can improve the accuracy of the rate-encoded classifier across different training strategies.
>  Thus, our results improve the evaluation of rate-encoded classifier ( CIFAR-10 CLEAN: 79.55 (m=1) to 83.22 (m=10), CIFAR-100 CLEAN: 50.9 (m=1) to 55.27 (m=10)), which was not observed earlier. Fig 1(a) reports the improvement in accuracy of rate-encoded models with m.
>
> Further, as we can see in Fig 1(d), the accuracy of the constant encoded models quickly breaks down under small adversarial perturbations, while rate-encoded models continue to offer better robust accuracy. Thus, under the threat of adversarial attacks, one can find the rate-encoded models more suitable compared to their constant encoded counterpart.
>
> **In Tab.1, the author's comparative perspective is that the robustness of constant-encoded SNN under PGD (C) is weaker than that of rate-encoded SNN under PGD (R), so rate-encoded SNN is better. However, I noticed that the robustness of constant-encoded SNN is better than that of rate-encoded SNN under GN attacks, and at the same time, constant-encoded SNN maintains good defense against FGSM (R) and PGD (R). Therefore, it seems that the so-called superior robustness of rate-encoded SNN does not have a good generalization effect, and it cannot maintain good defense ability under different attacks, which is an obvious limitations.**
>
>  Ans: First, we would like to highlight that fgsm(R)/pgd(R) attacks are ineffective against constant encoded models, similar to the ineffectiveness of the fgsm(C)/pgd(C) attacks against the rate-encoded models. The reason behind this is discussed in detail to answer the questions from reviewer 77u8.
>
> The reported results in Table 1 with GN attack uses Gaussian perturbation: $x+\delta$, where $\delta \sim N(0,\sigma^2)$, with $\sigma= 8/255 \approx 0.031$. The better results of the constant encoded models against Gaussian noise attacks quickly break down as we increase the strength of the attacks, as computed in the table below. For example, at $\sigma=0.1$, CLEAN(R) with m=10, provides 21.77\% higher accuracy than CLEAN(C)+RAT.
>
>
>  |$\sigma$  |  0.05 | 0.1 | 0.2 | 0.3 | 0.4 | 0.5 |
>  | ----- | ------- | ------ | ------ | ------ | ------ | ------ |
>  | CLEAN(C) | 86.53 | 59.58 | 22.8 | 14.63 | 13.14 | 11.73 |
>  | CLEAN(C)+RAT | 84.3 | 60.3 | 26.01 | 16.01 | 12.66 | 11.26 |
>  | CLEAN (R), m=1 | 78.38 | 77.74 | 72.58 | 61.77 | 48.38 |	35.13 |
>  | CLEAN (R), m=10 | 83.13 | 82.07 | 77.65 | 66.26 | 51.14 | 36.66|
>
>
> We have now included the above table in the appendix of the paper. We would also like to emphasize that while comparing the empirical robust accuracy of two models, we compare the minimum accuracy obtained by a model across any attack, as some attacks can be weak for one model but strong for another. This can be further seen from Table 5 (recently included in the appendix following suggestions from NxbL), where we implement adversarial attacks using different back-propagation algorithms.

---

> > ### Comment · Reviewer_FaRA · 2023-11-22
> >
> > I have read the relevant responses from the authors and I think they have effectively addressed my conserns. Therefore, I choose to increase my rating to 8.

---

> > > ### Author Response · Authors · 2023-11-23
> > >
> > > We thank the reviewer for the kind evaluation of our work.

---

### Official Review · Reviewer_wpSa · 2023-10-30

**Soundness:** 2 fair
**Presentation:** 2 fair
**Contribution:** 2 fair
**Rating:** 6
**Confidence:** 3

**Summary:**

This work presents a certified robust training framework for SNN. They mainly consider the Bernoulli noise in input and bounded by l1-norm. During robust training, they adopt STE for BP.

**Strengths:**

Apply certification based robust training on SNN is an interesting topic.

**Weaknesses:**

1.	Based on my understanding, the bound is conducted through repeated sampling (m times), which is a statistic boundary instead of a rigorous boundary. Please make it clearly.
2.	The robust training framework is not clear. Based on my understanding, the robust training is more like adding noise to inputs repletely to conduct the boundary, which seems trivial.
3.	The labels in experiments are misleading, i.e. I did not find RAT in tables.
4.	Please discuss [1]

Liang, Ling, et al. "Toward robust spiking neural network against adversarial perturbation." Advances in Neural Information Processing Systems 35 (2022).

**Questions:**

See weaknesses for details

---

> ### Author Response · Authors · 2023-11-20
>
> **The bound is conducted through repeated sampling (m times), which is a statistic boundary instead of a rigorous boundary. Please make it clearly.**
>
> Ans: Following the randomized smoothing framework, we estimate the output of the smooth classifier g, and probabilities $\underline{p_a}, \overline{p_b}$, through Monte Carlo estimation as given in eqn. (9). As stated after theorem 1, these bounds are obtained with high probability. This was further discussed in section 3.3, where we point to the proofs for the confidence of the estimates as obtained by Algorithms 1 and 2.
>
> **The robust training framework is not clear. Based on my understanding, the robust training is more like adding noise to inputs repeatedly to conduct the boundary, which seems trivial.**
>
>  Ans: In simple words, the adversarial training framework trains the models on adversarial perturbed images $x+\delta$, along with the original label y. Here, the adversarial perturbation $\delta$ for each image $x$ is calculated separately for each image, with respect to the present model weight obtained after each epoch. In theoretical formulation, the adversarial training is given as a min-max problem, where the inner maximization is performed over image perturbation and the outer minimization is done over the model weights, as described in eqn. (16).
>
>  Algorithmically, adversarial training differs from vanilla robust training in the sense that the perturbed image $x+\delta$ is found through some adversarial attack (e.g., FGSM/PGD) computed on the present model predictions, approximating the inner maximization. In contrast, in the robust training framework, the perturbation $\delta$ is a noise, possibly independent of the image $x$.
>
> In Table 1, column GN implements robust training with Gaussian noise, while columns FGSM and PGD implement adversarial training with respective attacks. To our knowledge, we are the first to report adversarial training results with rate-encoded SNNs, which beats the prior adversarial training results with constant encoded SNNs, as reproduced in Table 1. Further, we show that the results of the rate-encoded classifiers can be improved with the better approximation of the smooth classifier (m=1 vs. m=10)
>
>
> **The labels in experiments are misleading, i.e. I did not find RAT in tables.**
>
>  Ans: For each dataset CIFAR-10 and CIFAR-100, we report 4 sets of results in Table 1. Namely, constant encoding, constant encoding with RAT, rate-encoding with m=1, and m=10. Under each set there are four training methods, namely CLEAN (training with clean data), GN (training with Gaussian noise), FGSM (adv. training with FGSM attack), PGD (adv. training with PGD attack)
>
> **Please discuss [1]**
>
> Ans: The work adopts the existing certified training framework of IBP-CROWN based on bound-propagation, to the Heaviside function used in the SNN network, dubbed as S-IBP and S-CROWN. It is an important line of work, which we have now included in the related work section of the paper.

---

> > ### Comment · Reviewer_wpSa · 2023-11-22
> > **Response to authors**
> >
> > Thanks for the detailed response to answer my concerns. Considering the authors' responses and the robust training framework, I acknowledge the valuable contribution of this work. The formulation of noise in Theorem 5 is particularly commendable, and I am inclined to raise my score.

---

> > > ### Author Response · Authors · 2023-11-23
> > >
> > > We thank the reviewer for the kind evaluation of our work.

---

### Official Review · Reviewer_NxbL · 2023-10-31

**Soundness:** 3 good
**Presentation:** 3 good
**Contribution:** 2 fair
**Rating:** 5
**Confidence:** 4

**Summary:**

Firstly, the author established a connection between rate coding and randomized smoothing, theoretically providing robustness guarantees against adversarial perturbations under the l1 norm. Additionally, a novel adversarial training method was introduced for rate coding, significantly enhancing state-of-the-art empirical robust accuracy results.

**Strengths:**

1. The author establishes a connection between rate coding and randomized smoothing, providing a robustness proof for adversarial perturbations under the l1 norm.
2. The introduction of a new adversarial training strategy for rate coding notably enhances the state-of-the-art empirical robust accuracy.

**Weaknesses:**

The connection between encoding and robustness has indeed been explored in many previous studies, as evidenced by the references provided:
1.	"Rate Gradient Approximation Attack Threats Deep Spiking Neural Networks" from CVPR 2023
2.	"HIRE-SNN: Harnessing the Inherent Robustness of Energy-Efficient Deep Spiking Neural Networks by Training with Crafted Input Noise" from ICCV 2021
3.	"Rate Coding Or Direct Coding: Which One Is Better For Accurate, Robust, And Energy-Efficient Spiking Neural Networks?" from ICASSP 2022
4.	"Spike timing reshapes robustness against attacks in spiking neural networks" from Neural Networks
Given the substantial amount of existing work on this topic, it does seem that the author's contributions might not be as groundbreaking. Additionally, some studies, like [1], have indicated that SNNs encoded with Rate Coding using LIF neurons can be vulnerable. This further suggests a need for more innovative approaches or a unique perspective to truly stand out in this field.

**Questions:**

1.	If the author believes that rate coding is crucial for the robustness of SNNs, then it is essential to conduct experiments on converted SNNs and provide a comprehensive theoretical analysis. This will not only strengthen the validity of the claim but also offer insights into how rate coding impacts the robustness across different SNN architectures.
2.	The author should indeed expand the literature review to include more recent works related to SNN's attack and defense mechanisms. Relying solely on PGD and FGSM limits the scope and depth of the study. By incorporating a broader range of attack methodologies, the author can present a more holistic view of SNN's vulnerabilities and strengths in the face of adversarial attacks.

---

> ### Author Response · Authors · 2023-11-20
>
> We thank the reviewer for the detailed comments on the paper. We have carefully considered the suggestions and tried to incorporate them. We acknowledge the reference to the existing literature and promise to add them to the future version of the paper. We agree that the connection between encoding and robustness has been explored in various studies. However, we would like to emphasize that our work goes beyond the empirical observations of robustness and provides a provable robustness guarantee for rate-encoded SNNs against adversarial perturbations. We would like to briefly discuss the referred works:
>
>
>
>
> **RGA[1]** explores the rate-encoded nature of spikes that govern information propagation within SNN layers and argues that the rate of spikes is sufficient to encode the information. Following your recommendation, we have incorporated the RGA back-propagation technique, along with its close cousin called Backward Pass Through Rate (BPTR) as given in SNN-RAT(Ding et al., 2022). Thus, for each adversarial attack, we now have three implementations, namely, BPTT, BPTR, and RGA, reported in Table 5, which is now updated in the appendix of the paper.
>
> **HIRE-SNN[2]** explore specially crafted input noise to the constant coded inputs, where the noise is updated within the time-steps. We did not compare with this work directly, for (i). the robust training results were demonstrated on constant-encoding
> (ii). Under constant encoding, SNN-RAT (Table 4 of their paper) shows superior performance against adversarial attacks. However, we refer to their work in our related work section as they compare the spiking activity of constant and rate-encoded SNNs.
>
> The work **Rate Coding Or Direct Coding**[3] compares the accuracy of rate-encoded and constant encoded SNN trained with clean images under FGSM and PGD attacks and show superior accuracy of rate-encoded SNNs under different attack radius (Fig. 3 of their paper). This is consistent with our claims of superior robustness of rate-encoded SNNs. Our experimental contribution adds to their work (i). by including adversarial training of (smooth) rate-encoded classifier, (ii). by showing a smooth approximation of rate-encoded SNNs can significantly improve the performance of rate-encoded SNNs.
>
>
>
> **Analysis of converted SNNs** The theory of randomized smoothing holds irrespective of the base classifier that we choose. That is, whether we use the base classifier $f$ (in eqn. 6) from an adversarially trained model or a converted model, the corresponding smooth classifier g, will be robust against adversarial perturbation, i.e., $g(x+\delta) = g(x)$. However, if the base classifier is unfit, the prediction $g(x)$ can be incorrect, leading to poor clean and robust accuracy.
>
> With your suggestion, we conducted experiments with ANN-to-SNN converted VGG-16 models on the CIFAR-10 dataset using a recent conversion technique [Bu et. al. 2021]. We find that under rate-encoding, converted SNNs do not have good clean accuracy at small latency T, which is rather a limitation of the base classifier trained to work with direct inputs, without Bernoulli noise. This can even happen to constant encoded SNN trained on clean images if we add large Gaussian noise to the input (please see the table provided in reply to reviewer  FaRA )
>
> **Constant Encoding**
>  |Attack|  T=4|  32|  64| 128 |
> | ----- | ----- | ----- | ----- | ----- |
>  | clean|  92.58|  95.47|  95.47| 95.51 |
>  | gn |  85.84|  89.1|  88.98| 88.97 |
>  | fgsm(C)|  21.02|  14.25|  15.85| 17.56 |
>  | pgd(C)|  **0.2**|  **0.03**|  **0.02**| **0.03**|
>  | fgsm(R)|  73.59|  64.54|  37.34| 14.7|
>  | pgd(R)| 87.46| 82.53| 58.36|11.9 |
>
>
> **Rate Encoding**
>  |Attack|  T=4|  32|  64| 128 |
>  | ----- | ----- | ----- | ----- | ----- |
>  | clean| 11.83| 24.12| 41.5|63.11|
> | gn | 11.54| 22.8| 38.35|57.4|
> | fgsm(C) | 11.14| 17.43| 23.64|28.8|
>  | pgd(C) | 11.24| 17.56| 23.54|26.2|
>  | fgsm(R) | 11.49| **16.36**| **18**|**15.76**|
>  | pgd(R) | 11.46| 17.54| 18.94|13.92|
>
>  It is interesting to note that, while the clean accuracy of rate encoding is lower than that of directly trained SNNs, the robust accuracy (minimum accuracy across among the attacks) of rate encoding surpasses that of SNNs with constant encoding.
>  (ii). the clean accuracy of rate encoding keeps improving larger T, but the robust accuracy initially increases (due to better prediction) but eventually drops, possibly due to the effect of T as found in our theory. (T=4: 11.49, T=32:16.36, T=64: 18, T=128: 15.76)
>  A similar observation was made in directly trained SNNs, as given in Table 4 in the appendix, where we evaluated the models at T=4 vs T=8.
>  (iii) A similar drop in robust accuracy for constant encoded models may hint at the rate-encoded nature of spikes in general SNN layers, as reported by [1].
>
> We have now included these results in Table 7 in the appendix of the paper.

---

> > ### Author Response · Authors · 2023-11-23
> >
> > We would like to request the reviewer to please share their evaluation of the answers.

---

### Official Review · Reviewer_77u8 · 2023-10-31

**Soundness:** 3 good
**Presentation:** 2 fair
**Contribution:** 3 good
**Rating:** 8
**Confidence:** 3

**Summary:**

The manuscript investigates the adversarial robustness of spiking neural networks (SNN). It establishes a connection between rate-encoding used for SNNs and the randomized smoothing framework from adversarial robustness. It clarifies why rate-encodings are more adversarial robust than constant-encodings, as observed previously in empirical studies. The established theory confirms the empirical observation that this benefit diminishes with increased SNN inference latency. In addition, empirical robustness is investigated using a variety of adversarial training techniques optionally combined with randomized smoothing for both constant and rate encodings.

**Strengths:**

* **Original Contribution:** The paper offers a unique perspective on adversarial robustness for SNNs using rate encoding. The connection between rate encoding and randomized smoothing is a novel and interesting insight contributing to the field.
* **Theoretical work**: The authors provide a theoretical foundation for the robustness of rate-encoded SNNs. While the results rely on previous work, the extension to SNNs is novel and non-trivial (lemma 3 and 4).
* **Comprehensive empirical results**: Additionally to the theoretical contributions, the authors combine their results with various adversarial training settings to improve empirical robustness.
 * **Clarity of introduction and background:** The paper is well-structured and provides clear explanations of the relevant concepts, making it accessible to a wide audience, including those not specialized in spiking neural networks or adversarial robustness.

**Weaknesses:**

## Major issues
* **Claims and limitations**: The manuscript claims to "introduce a novel adversarial training algorithm ... improves state-of-the-art ...". Yet it is not very clear what the novelty exactly is (e.g. as also stated by the authors Stalman et al. 2009 did combine adversarial training and randomized smoothing. Is it the adaption to SNNs? The modifications required for the attack?). This should be more clear. Further, the manuscript claims new state-of-the-art performance in the abstract. But only "compare the performance of the proposed methods with state-of-the-art adversarial training algorithms" in the Experiments. These are two slightly different statements, and they should be clearly formulated (i.e. what is state-of-the-art (is there a well-established state-of-the-art in the field?), and exactly which proposed method beats it?). While the authors do discuss the obtained certified radius in section 5.3, these certifications seem way smaller than for the ANN case [2]. While a direct comparison is not necessary, this should be discussed more explicitly.

* **Empirical details and clarity of experimental results**: The manuscript lacks a more comprehensive description of experimental details, which should be added to the appendix i.e., how many PGD iterations, all the training details for all adversarial training methods, the cost of training with the different methods (GPUs, time), … . There also is only a brief discussion of the empirical results, and some of the results are unclear/surprising to me (see questions). This should be discussed in more detail.

* **Comparison between certified and empirical robustness**: The manuscript does analyze both the empirical and certified robustness. Yet, does treat them mostly independently. It would be helpful to e.g. take Fig 1 (d) (i.e. the empirical robustness for various epsilon) and also plot the certified robustness (with radius r=epsilon). This information is already displayed in Table 3 (although for a much smaller radius). This would nicely showcase the gap between empirical and certified robustness. While the authors do not claim tightness of the certifications, this at least should be discussed/visible.



## Minor issues and recommendations:

* The authors state that certified defense methods based on convex relaxations and branch and bound are restricted to ReLU activations. This is incorrect as [1], does generalize to general activation functions. The claim that these methods do not apply to SNNs may still hold and should be checked by the authors.
* Some of the proofs rely on the differentiability of g with respect to x. It might be good to point out that g is differentiable with respect to x, even if f is not differentiable with respect to z (as is the case for SNNs, just for clarity).
* There are some language errors (e.g. page 2 row 25) and formatting errors (whitespace in front of citations e.g. page1 first paragraph last row, page 2 row 11, ...).




[1] Formal Verification for Neural Networks with General Nonlinearities via Branch-and-Bound  (Zhouxing Shi et. al.)

[2] https://sokcertifiedrobustness.github.io/leaderboard/

**Questions:**

Multiple open questions about results in Table 1, for example:
* Why do fgsm(R) and pgd(R) attacks not work on models trained with constant encoding? As fgsm(C) and pgd(C) do work well as expected, shouldn't this indicate that rate-encoding messes up the attack algorithms (i.e., because the gradients become noisy)?
* As one tests attacks on both encodings (C/R), it might be good also to have clean (C/R), no? (it is rather interesting to me that models trained on C, do work that well on fgsm(R) and pgd(R) and visa versa)
* Adversarial training is performed both for FGSM and PGD. As PGD is the “stronger” adversary, I would expect it to be more robust, yet it is not. Especially, the PGD-adversarially trained network performs worse than an FGSM-adversarially trained network if tested against a PGD adversary. This result seems counterintuitive and is consistent across different datasets. Why? What PGD adversary was employed during training (as the evaluation PGD is as expected better than FGSM) ?

---

> ### Author Response · Authors · 2023-11-20
>
> We are grateful to the reviewer for detailed review and insightful comments. We list our answers below:
>
>
> **Specify algorithmic novelty:** As suggested by the reviewer, we now mention in the paper, that the algorithm novelty lies in adapting the randomized smoothing technique for rate-encoded SNN. More concretely,
> (i) Table 1, improves the accuracy of rate-encoded classifiers through better approximation of the smooth classifier, i.e., we could observe results with m=10, are better than m=1.
> (ii) Adversarial training of rate-encoded classifier, by employing rate-encoded adversarial attacks are also reported first time to our knowledge.
>
> **State-of-the-art:** We intend to say the state-of-art in adversarial training, we compare our results with another existing adversarial training algorithm, SNN-RAT, and show improvement in robust accuracy (i.e., minimum empirical accuracy found under any attack). (FGSM(C):39.75, FGSM(C)+RAT: 42.76 vs. FGSM(R): 51.63)
>
> **Comparison of certified and empirical robustness:** We agree with the reviewer to discuss this gap in detail in the paper. However, we would like highlight, that the certified radius for robustness in Table 3 is obtained under $l_1$-norm, while figure 1(d) reports attack radius under $l_\infty$-norm, as customary for the FGSM/PGD attacks. Thus, these results can not be plotted together unless we perform some bound transfer for the certified radius.
>
> **Training Details:** As suggested by the reviewer, we have added all the training hyper-parameters/details in the appendix. We answer more specific questions next.
>
> **Why do fgsm(R) and pgd(R) attacks not work on models trained with constant encoding?**
>
> Ans: We would like to clarify how fgsm(R) works on a constant encoded model. To find the perturbation $\delta$, we use rate-encoding of $x$, compute the loss, perform the gradient ascent step, and find $\delta$. Now the perturbed image stands to be $x+\delta$, which we evaluate with constant encoding. Thus, $C/R$ here stands for the encoding used to compute the attack,
> however, once the attack is found, the perturbed input is evaluated with default model encoding.
>
> We observe, when a model is trained on a particular encoding and the attack also uses the same encoding, the attacks are more effective in fooling the model parameters. As $\delta$ found in this process is not informative enough, fgsm(R)/pgd(R) attacks are not able to find effective perturbation on a constant encoded model. And similarly, fgsm(C)/pgd(C) do not work well on the rate-encoded models. Please see this answer in conjunction with the next answer.
>
>
>
>
> **As one tests attacks on both encodings (C/R), it might be good also to have clean (C/R).**
> Ans:  As clarified above, although the attack fgsm(R) uses rate encoding to find $\delta$, the perturbed image $x+\delta$ is evaluated using the model encoding. Thus, while evaluating a clean image, since there is no attack involved, we chose the same encoding as the model encoding.
>
> However, as the reviewer suggests, we can always evaluate a model trained with constant encoding with rate encoding, and vice versa. But as the true model weights are not trained on these inputs, the clean accuracies are not good enough. We report the accuracies below:
>
>
>
> **CIFAR-10, Train: Constant Encoding**
> |               | CLEAN  | GN | FGSM(C)| PGD(C) |
> | ----------- | ----------- |----------- |----------- |----------- |
> | clean(C)  | 92.15 | 91.7 | 79.4 | 79.15 |
> | clean(R)   | 13.93 | 20.01 | 11.55 | 19.88 |
>
> **CIFAR-10, Train: Rate Encoding**
> |               | CLEAN  | GN | FGSM(R)| PGD(R) |
> | ----------- | ----------- |----------- |----------- |----------- |
> | clean(R)  | 79.55 | 79.36 | 76.89 | 76.36 |
> | clean(C)  | 12.01 | 15.66 | 39.11 | 13.63 |
>
> The results further explain the reason fgsm(R) attack does not work well with constant encoded models, as to compute the perturbation, we should start with output which is correct. As the clean accuracy with the opposite encoding is itself not very good, the perturbations found by the attack are not effective.
>
>  However, the result that rate-encoding does not work on a model trained with constant encoding should not be seen as a limitation of the rate-encoding technique. The randomized smoothing technique only guarantees that the prediction of the smooth classifier will not change upon perturbation of the input, which does not require that the prediction of the smooth classifier in the first place is correct. To obtain robustness along with correct prediction, we need to supply a good base classifier, such as the different classifiers (adversarially) trained with rate encoding.

---

> > ### Author Response · Authors · 2023-11-20
> >
> > **As PGD is the “stronger” adversary, I would expect it to be more robust, yet it is not.**
> >
> > Ans: Thank you for highlighting this issue. We observed that in adversarial training with PGD attacks, the PGD attacks were not powerful enough compared to testing. Only 2 pgd iterations were used in training (with $\eta = \frac{1}{255}$, $\epsilon= \frac{8}{255}$, from eqn. (4)), while testing uses 7 pgd steps. We increased the number of pgd steps to 4 in training and set  $\eta = \frac{2}{255}$, $\epsilon= \frac{8}{255}$ , and recomputed the results for all adversarial training with PGD, which solved the anomaly. The new results are now updated in Table 1.
> >
> > **Minor Issues**
> >
> > We have added references to the recent works involving certified training with bound propagation.
> >
> > We would highlight the differentiability of g, irrespective of the differentiability of f.

---

> > > ### Comment · Reviewer_77u8 · 2023-11-22
> > >
> > > I thank the authors for the efforts you put into addressing the concerns raised in my review. I appreciate the clarifications and additional details provided in your response.
> > >
> > > With the provided clarifications and updated results, the empirical results are now in agreement with my expectations. The authors addressed my concerns, and hence, I tend to raise my rating.
> > >
> > > I think one could still include a better comparison between certified and empirical robustness. While I acknowledge that the current results are based on different norms, hence hard to compare. It is rather easy to modify both FGSM and PGD to work with L1-norms (but would require additional work). The revised manuscript did introduce some formatting issues i.e., no white space between text and citation or no brackets at all (e.g., at the end of the introduction, …). This should be fixed for the final version.

---

> > > > ### Author Response · Authors · 2023-11-23
> > > >
> > > > We sincerely thank the reviewer again for the helpful suggestion on the comparison. We will try to include such a result in the final version of the paper. Presently, we have fixed the formatting errors and reuploaded the draft.

---

### Meta-Review · Area_Chair_RmB2 · 2023-12-05

**Metareview:**

The paper studies the adversarial robustnesss of spiking neural networks-- it describes a connection between the rate-encoding often used for spiking networks, and randomised smoothing for adversarial robustness, and uses it to explain by rate-encoding schemes are often found to be adversarially robust.

The reviewers agreed that the paper was sound and novel, and that the combination of theoretical results and empirical results was convincing. There were concerns about clarity and details on experimental results, and description of related work.

**Justification For Why Not Higher Score:**

If anything this is a bit of a borderline paper, certainly not higher than 'accept'.

**Justification For Why Not Lower Score:**

I would be ok with the paper being down-weighted-- however, the reviewers agreed on the novelty of the result, and appreciated the new connection between randomised smoothing and rate encoding. I see no reason to overrule their consensus [one reviewer gave a 5, but they also did not participate in the discussion, despite reminders by the authors], even if I am not sure how much of an impact this paper will have, they do seem to study a somewhat specialised problem.

---

### Decision · Program_Chairs · 2024-01-16

Accept (poster)